# MODEL MERGING BEYOND IMAGE CLASSIFICATION: A REALITY CHECK

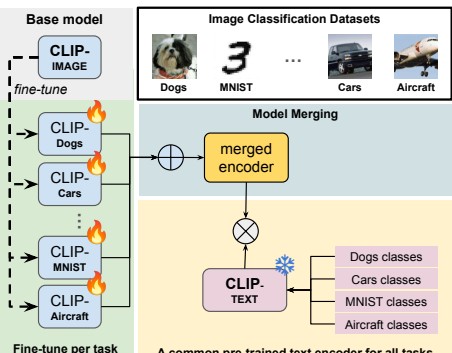 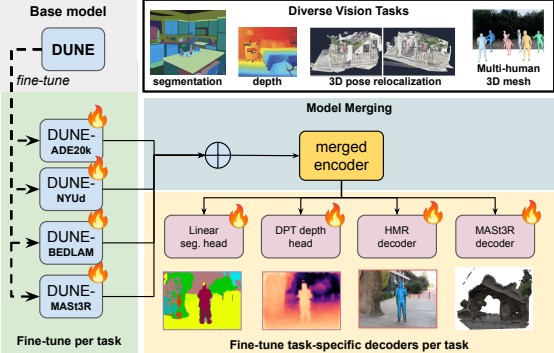

Figure 1: **Model merging in computer vision.** *Left:* Existing benchmarks evaluate model merging on CLIP-based image classification, where merged encoders are simply combined with an out-of-the-box *frozen* text encoder for all tasks. Hence, model evaluation is fast and an exhaustive ablation of merging hyper-parameters is possible. *Right:* This paper focuses on challenging heterogeneous tasks from DUNE (Sarıyıldız et al., 2025) so encoders need to be combined with *trainable* task-specific decoders, making hyperparameter search extremely inefficient.

## ABSTRACT

Efficiently merging several models fine-tuned for different tasks, but stemming from the same pre-trained base model, is of great practical interest. Despite extensive prior work, most evaluations of model merging in computer vision focus on image classification using CLIP, where different sets of labels define different classification tasks. This paper ventures model merging into the more challenging setup where the different tasks operate in different output spaces and thus rely on different *trainable* decoders. This renders exhaustive hyperparameter search impractical. To address this, we introduce the *task alignment score*, and show how it can be used to i) speed up hyperparameter selection by orders of magnitude and ii) generalize proxy-based hyperparameter selection to any task. We also find that, in our setting, several recent models fail and show it is largely due to some fine-tuned models being significantly further-away from the base than others, leading to a strong unbalance. More importantly, we demonstrate that, thanks to our contributions, model merging remains effective and can improve the performance of a state-of-the-art multi-task vision model.

## 1 INTRODUCTION

Vision foundation models – *i.e.* large models trained on gigantic amounts of data – are generic encoders that can be used out-of-the-box in many downstream tasks (Oquab et al., 2024; Alayrac et al., 2022; Sarıyıldız et al., 2025; Ranzinger et al., 2024; Radford et al., 2021) – typically in conjunction with smaller task-specific decoders. Fine-tuning these models on task-specific data yields higher accuracy, however, deploying several fine-tuned encoders becomes costly, as each adds memory and compute overhead. This is especially problematic in constrained settings such as robotics, where localization, human detection, and semantic or geometric scene understanding must

run simultaneously on limited hardware. This raises the question: can we *efficiently* merge these specialized encoders into a single one, retaining performance while lowering deployment cost?

The literature on model merging explores this question and several works (Ilharco et al., 2023; Lee et al., 2025; Ortiz-Jimenez et al., 2023; Wang et al., 2025) have shown that it is possible to merge several models fine-tuned from the same base model without additional training, by simply performing "arithmetic" in the weight space. This paper focuses on such *arithmetic-based merging* methods in the realm of computer vision, and, unless otherwise stated, *model merging* refers to this particular setting.

Although merging methods have seen remarkable progress, their *evaluation in computer vision has been so far limited to CLIP-based image classification tasks* (see Fig. 1, left).

Concretely, we extend model merging to a broader range of vision tasks that share the same encoder but require task-specific decoders (see Fig. 1, right). In this context, model merging is only applied to the visual encoder and we cannot assume that all tasks share the same fine-tuning setup and decoder anymore. This makes task vectors (weight differences between fine-tuned and base models) highly imbalanced, and the evaluation of merged models becomes more costly, severely altering the performance and computational costs of recent methods as we show in our experimental study.

For our experiments, we adopt the heterogeneous task benchmark from DUNE (Sarıyıldız et al., 2025), which covers four different 2D and 3D tasks: semantic segmentation, depth estimation, 3D human mesh recovery, and 3D reconstruction. We use DUNE as our *base model*, given its state-of-the-art performance. In this set-up, evaluating merged models is extremely costly, as performance depends on task decoders fine-tuned after merging, rendering exhaustive hyperparameter search impractical (see Fig. 2).

To address this, we introduce the *Task Alignment Score* (**TAS**), a proxy performance score that measures the feature distance between the merged and fine-tuned models. TAS is cheap to compute, correlates well with actual performance, and enables methods like AdaMerging (Yang et al., 2024)–previously limited to classification–to be applied in this setting. We also experimentally show TAS can be used as a simple and generic proxy for performance by any merging method to significantly reduce the cost of hyperparameter search–making it practical in this setting.

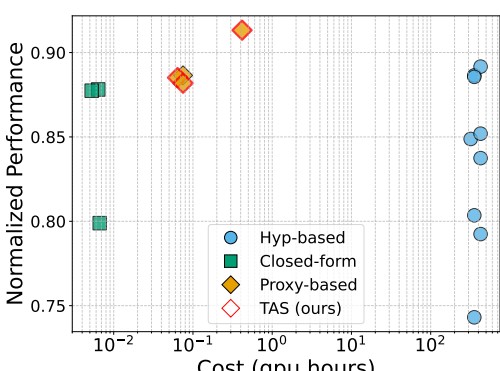

Figure 2: **Model merging for realistic tasks.** Each marker is a method. Hyperparameter-based methods are too costly to be used in practice. When combined with our proposed TAS, existing methods become both more competitive and much more efficient in this context (*cf.* Tab. 1).

To summarize, our contributions are as follows: **i)** We extend model merging evaluation beyond classification and reveal key assumptions in prior setups that do not hold in the general setting; **ii)** We propose TAS, a general and efficient proxy to guide hyperparameter search and leverage it to apply merging methods to more challenging and diverse tasks; **iii)** We show that merging can improve state-of-the-art multi-task models, with several merging methods improving the (already competitive) base model performance. To the best of our knowledge, this work is the first one to study model merging on such a challenging and diverse set of vision tasks.

## 2 BACKGROUND AND RELATED WORK

**Preliminaries and notations.** Let $\theta_0 \in \mathbb{R}^d$ denote the parameters of a pretrained visual encoder, referred to as the *base model*. We assume that this base model is fine-tuned end-to-end independently for each task $t$ in a set of $T$ downstream tasks. Unlike the standard zero-shot classification setting, fine-tuning for diverse vision tasks typically involves appending task-specific *decoders* to the shared encoder and optimizing task-specific loss functions.

Let $\theta_t$ represent the encoder parameters after fine-tuning for task $t$, defined as $\theta_t = \arg\min_\theta \mathcal{L}_t(\theta)$ where $\mathcal{L}_t(\theta)$ is the training loss. Following Ilharco et al. (2023), we define the *task vector* (TV) for task $t$ as $\tau_t = \theta_t - \theta_0$. The goal of model merging is to find a single set of encoder parameters that jointly minimizes the losses across all tasks, *i.e.* $\theta^* = \arg\min_\theta \sum_{t=1}^T \mathcal{L}_t(\theta)$. *Arithmetic-based merging* methods aim to efficiently approximate $\theta^*$ by combining task vectors in the parameter space. Most merging methods can be framed as a variation of the following weighted sum:

$$\theta_{\text{merged}}(\lambda, \mu) = \theta_0 + \sum_{t=1}^T \lambda_t \odot \phi(\tau_t; \mu), \tag{1}$$

where $\lambda_t \in \mathbb{R}^d$ is a vector of task-specific weights, $\phi : \mathbb{R}^d \to \mathbb{R}^d$ is a transformation parametrized by hyperparameters $\mu$ and $\odot$ is the Hadamard product. The simplest case corresponds to uniform model averaging, where $\lambda_t = [1/T, ..., 1/T]$ and $\phi$ is the identity function. Let $\lambda$ denote the set of all $\lambda_t$ parameters. When there is no ambiguity, we denote $\theta_{\text{merged}}(\lambda, \mu)$ as $\theta(\lambda, \mu)$ or just $\theta$.

**Closed-form methods** compute the merged model directly from the weights, without using any data during merging. These are the most efficient merging methods. The simplest is **model averaging** (Matena & Raffel, 2021; Wortsman et al., 2022; Jin et al., 2023; Choi et al., 2024), which is also the analytical solution to the merging objective under the assumption that (i) each fine-tuned model is optimal for its task and (ii) performance is inversely proportional to the euclidean distance between merged and fine-tuned models in weight space. **MetaGPT** (Zhou et al., 2024) makes task weights proportional to the $\ell_2$ norm of task vectors, a refinement derived from minimizing loss-distance under assumptions of linearity and orthogonality of task vectors.

**Hyperparameter-based methods** treat $\lambda_t$ and, when applicable, $\mu$ as hyperparameters to be selected based on downstream task performance. **Task Arithmetic** (Ilharco et al., 2023) extends weight averaging by linearly interpolating between the average of fine-tuned models and the base model. In **TIES** (Yu et al., 2023), $\phi$ applies weight pruning to task vectors, retaining the top-$k$ values with highest magnitudes and consistent signs, in order to reduce task interference. **Breadcrumbs** (Davari & Belilovsky, 2023) also applies pruning, but removes both high- and low-magnitude weights at the layer level to eliminate outliers. **Consensus** (Wang et al., 2024) prunes weights per task, then constructs a consensus mask by retaining weights that appear in more than one task-specific mask. **Lines** (Wang et al., 2025) uses per-layer weights where coefficients increase linearly across layers, based on the intuition that early layers capture general features (thus closer to the base model) whereas later layers are more task-specific.

More recently, several methods have used SVD-based compression per layer, when weights are matrix-shaped (defaulting to Task Arithmetic otherwise). For instance, **STAR** (Lee et al., 2025) performs SVD truncation on each task vector, retaining the top singular values required to preserve a specified fraction of the energy, and rescales the remaining singular values to match the initial energy. **TSV** (Gargiulo et al., 2025) applies SVD twice: first independently to each task vector (as in STAR), and then again after concatenating the decomposed matrices across tasks to enforce orthogonality. It has two flavors: with SVD truncation (lossy) and without (lossless).

Finding the optimal merging hyperparameters, such as the lambdas or the pruning and SVD truncation thresholds, although efficient for CLIP-based classification (Fig. 1, left), becomes prohibitively expensive when tasks require *trainable* decoders for evaluation, as is common in computer vision (Fig. 1, right). In Sec. 3, we introduce the task alignment score proxy and show it can be used to accelerate hyperparameter-based methods by several orders of magnitude.

**Proxy-based methods** aim to efficiently approximate task performance without access to labels, thereby enabling faster selection of merging hyperparameters. **AdaMerging** (Yang et al., 2024) minimizes entropy as a proxy objective, inspired by test-time adaptation. Merging coefficients, such as per-task $\lambda_t$, are then optimized via gradient descent. However, entropy minimization can only be applied for categorical tasks *e.g.* classification. **AdaMMS** (Du et al., 2025) was designed to merge LLMs and selects merging weights based on generation consistency between models obtained with adjacent hyperparameter values. Such consistency requirement restricts it to methods with a single scalar hyperparameter. In Sec. 3, we propose an alternative proxy to entropy minimization or generation consistency, that extends AdaMerging and AdaMMS to *any* task.

**Alternative model merging applications.** Recently, some studies have explored merging in the context of continual learning. Dziadzio et al. (2025) use CLIP-based tasks in a continual learning setting where training data and inference objectives evolve over time. They explore various merging and initialization strategies to improve downstream performance. Sokar et al. (2025) investigate model merging to mitigate catastrophic forgetting when fine-tuning Vision Language models on a continual VQA benchmark. Interestingly, although our setup differs significantly, these studies also find that simple methods (such as weight averaging or task arithmetic) perform surprisingly well, often matching or outperforming more complex approaches. To the best of our knowledge, no prior work has evaluated model merging on a set of diverse vision tasks spanning both 2D and 3D settings.

## 3 Model merging beyond CLIP-based image classification

As discussed before, most model merging evaluations in vision have focused on CLIP-based zero-shot image classification across different datasets. In this section, we ask: *Can we apply model merging to any set of vision tasks?* We first discuss the new challenges introduced when extending model merging beyond classification (Sec. 3.1). In particular, key assumptions from classification-based settings – such as cheap evaluation, the applicability of entropy-based proxies, and the feasibility of exhaustive hyperparameter tuning – no longer hold. This motivates our generic and efficient performance proxy to efficiently guide hyperparameter selection in more complex scenarios (Sec. 3.2).

### 3.1 Implicit assumptions of CLIP-based image classification

The dominant evaluation setup for model merging in vision builds on CLIP's zero-shot classification capabilities (see Fig. 1, left). In this setup, the visual encoder is fine-tuned separately for each task, while the frozen text encoder serves as a shared classifier (*i.e.* a very light-weight decoder) by embedding class names or descriptions (Radford et al., 2021; Ilharco et al., 2021). As a result, the visual encoder effectively constitutes the *entire model* used to solve all tasks.

This setup has led to two implicit assumptions in the model merging literature: (1) *evaluation of merged models is cheap*, requiring only forward inference; and consequently (2) *exhaustive search over merging hyperparameters is feasible*. As a result, many recent methods rely heavily on such search to select optimal merging configurations (see Sec. 2).

However, these assumptions do not hold in general when merging models across different vision tasks. In such standard multi-task setups, models typically consist of a shared encoder and multiple task-specific decoders. Merging is only performed on the encoder, and each task requires fine-tuning or re-training its decoder after merging in order to achieve competitive performance. This makes the evaluation of hyperparameter-based merging methods significantly more expensive, since evaluating each merged model on every task now involves retraining (potentially large) decoders.

One way to bring this setup closer to the CLIP-based formulation would be to fine-tune only the encoders, keeping decoders frozen. However, we experimentally found that encoder performance with frozen decoders does not correlate well with performance with trained decoders (see Appendix C). Another possible solution is to use *proxy* signals to estimate downstream performance without performing an expensive retraining. Proxy-based approaches such as AdaMerging (Yang et al., 2024) and AdaMMS (Du et al., 2025) follow this direction. However, AdaMerging relies on entropy minimization, which is only applicable to categorical tasks such as classification, while AdaMMS is specifically designed for generative models and assumes one can compare generated outputs. These limitations motivated us to introduce a more generic proxy measure, applicable across *any* task, enabling both efficient hyperparameter selection and learning-based merging.

### 3.2 The Task Alignment Score

Motivated by the prohibitive cost of hyperparameter tuning when tasks require trainable decoders, we propose a generic proxy that quantifies the performance drop incurred by swapping, for a given task $t$, the fine-tuned parameters $\theta_t$ with the merged model $\theta_{\mathrm{merged}}$. We refer to this proxy as the **task alignment score** (**TAS**). The intuition behind TAS, which is reminiscent of work on model distillation, is the following: if the encoder output remains unchanged when the task-specific parameters $\theta_t$ are replaced by the merged model parameters $\theta_{\mathrm{merged}}$, then the performance of both models should be

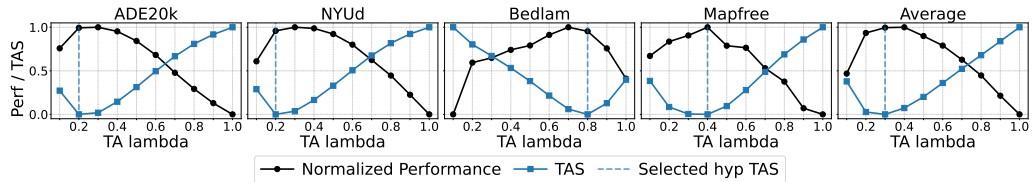

Figure 3: **TAS vs. performance** for different values of the merging coefficient $\lambda$ when using Task Arithmetic (TA). The selected value is indicated by a dashed line.

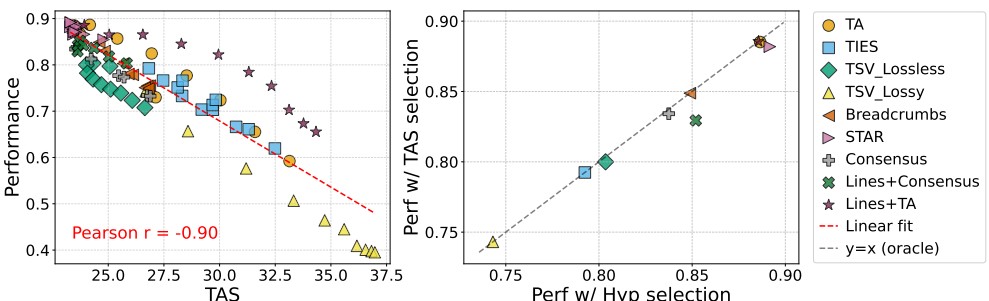

Figure 4: **Left**: Average normalized performance across tasks for different merging methods and hyperparameter settings *vs.* the average TAS score. **Right**: Performance of the models selected with TAS *vs.* the actual best performance when exhaustively evaluating all hyperparameter configurations. All methods are close to $y = x$, validating TAS as a proxy for evaluation performance.

similar for this task. Computing TAS only requires access to *unlabeled* data from each downstream task and is both task-agnostic and computationally efficient, as it only involves inference.

Formally, let $D_1, \ldots, D_T$ denote unlabeled training image set for each task. Let $f(x; \theta) : \mathcal{X} \to \mathcal{F}$ denote the image encoder parameterized by $\theta$, where $\mathcal{X}$ is the input space and $\mathcal{F}$ the output feature space, and let $x \in \mathcal{X}$. Given a dissimilarity function $d : \mathcal{F} \times \mathcal{F} \to \mathbb{R}^+$, we define the *task alignment score* for task $t$ as the average dissimilarity between features produced by the merged model and the task-specific fine-tuned model over all samples in $D_t$:

$$\text{TAS}_t(\theta) = \frac{1}{|D_t|} \sum_{x \in D_t} d\big(f(x; \theta), f(x; \theta_t)\big), \tag{2}$$

To obtain a single score across all tasks, we compute the average: $\text{TAS}(\theta) = \sum_{t=1}^{T} \text{TAS}_t(\theta)/T$.

**Is TAS a good proxy for performance?** In Fig. 3 we show TAS for each task across different hyperparameter values *vs.* the actual performance after the costly per-task decoder fine-tuning, for the original Task Arithmetic method (see Sec. 4 for details). We observe a strong negative correlation for each task and for the average across all tasks. In Fig. 4 (left) we show the average normalized performance across tasks for different merging methods and hyperparameter settings *vs.* the average TAS and also observe a strong negative correlation. Moreover, in Fig. 4 (right) we show the performance of the models selected with TAS *vs.* the actual best model for each method found via exhaustive evaluation of hyperparameter configurations. We observe that all methods are close to the $y = x$ line, corresponding to always selecting the best performance.

In the following sections, we combine TAS with some of the methods to select hyperparameters. We denote such combinations with an additional "+TAS" (*e.g.* AdaMerging+TAS). This is done by selecting the $\lambda$ (and whenever relevant $\mu$) parameters that minimize TAS, *i.e.* $\lambda^*, \mu^* = \arg\min_{\lambda,\mu} \text{TAS}(\theta_{\text{merged}}(\lambda, \mu))$, where $\theta_{\text{merged}}(\lambda, \mu)$ is given by Eq. (1). We note that the $\arg\min$ operation takes different meanings for different model merging approaches: it might involve an exhaustive search over the parameter/hyperparameter space or a learning-based procedure in the case of continuous optimization.

In Appendix E we show more plots of TAS *vs.* per-task and average performance for different methods. We ablate different distance metrics between features and find TAS to be very robust. We also observe TAS is extremely robust to the number of samples used to compute the set of features, which makes it very efficient. By default we use the $\ell_2$ distance and 128 samples per task.

# 4 EVALUATING MERGING METHODS UNDER DIVERSE VISION TASKS

## 4.1 EXPERIMENTAL PROTOCOL

**Base model and tasks.** We adopt the evaluation protocol introduced in DUNE (Sarıyıldız et al., 2025) to assess model merging across a diverse set of 2D and 3D vision tasks; we refer to it as the DUNE benchmark. Specifically, we evaluate on semantic segmentation using ADE20k (Zhou et al., 2019), depth estimation on NYUd (Silberman et al., 2012), 3D human mesh recovery on Bedlam (Black et al., 2023), and pose relocalization on Niantic's MapFree dataset (Arnold et al., 2022). As a base model we take DUNE (ViT-Base/14), given its demonstrated performance across a range of *diverse* tasks, precisely the setting we want to study.

**Fine-tuned models and decoders.** We fine-tune the base model end-to-end for each task, resulting in four task-specific encoders. After merging, we freeze the encoder and fine-tune only the corresponding task-specific decoders. For the decoders, we use a linear head for semantic segmentation (ADE20k) and a DPT model for depth estimation (NYUd), following Oquab et al. (2024). For human mesh recovery on Bedlam, we adopt the two-headed architecture from MultiHMR (Baradel et al., 2024). For visual relocalization on MapFree, we use a ViT-Large decoder as proposed in MASt3R (Leroy et al., 2024).

**Evaluation protocols.** Due to the high cost of model evaluation, we introduce two protocols: the *ablation* and the *long fine-tuning*. The *ablation protocol* is used for the experiments in Tab. 1, including, when applicable, hyperparameter ablations. In this protocol, we reduce the number of fine-tuning iterations for task-specific decoders; in the case of MapFree, also limit the amount of training data. Moreover we use a linear head (instead of DPT) for depth estimation. To avoid bias towards the original validation sets, evaluations are conducted on custom held-out sets. We restrict the total hyperparameter evaluation budget to a maximum of 12 configurations per method. This constraint ensures that all hyperparameter-based methods operate under comparable computational budgets. Only the best result per method is reported in Tab. 1. On the other hand, the *long fine-tuning protocol* follows exactly the evaluation of DUNE, so results are directly comparable to those in Sarıyıldız et al. (2025). It is used in Tab. 2.

**Implementation details and evaluation metrics.** Our experiments are based on the public DUNE codebase[1] and we use the official implementations of the merging methods whenever available. For each task, we report the standard evaluation metrics, along with a *normalized performance metric* that we define as $1/T \sum_{t=1}^{T} \rho_t(\theta_{\mathrm{merged}})/\rho_t(\theta_t)$, where $\rho_t$ is the relevant metric for task $t$ (*e.g.* mIoU) when higher is better, and when lower is better we use the inverse of the metric (*e.g.* $\rho_{\mathrm{NYUd}} = 1/\mathrm{rmse}$). This facilitates comparisons across tasks with different scales and metrics.

**Generalizing proxy-based methods with TAS.** Proxy-based methods such as AdaMerging and AdaMMS were originally developed for a specific set of tasks; classification and generation, respectively. We generalize them so they can work with encoders for *any* vision task using variations of TAS. For instance, we replace the original loss in AdaMerging with TAS to guide the optimization of merging coefficients; we refer to this variant as **AdaMerging$_{+\mathrm{TAS}}$**. We denote by **AdaMMS$^*$** a generalized variant of AdaMMS adapted beyond text generation, where instead of generation consistency, model distance is measured using $d(f(x; \theta_1), f(x; \theta_2))$ from Eq. (2). Further details are provided in Appendix H and Appendix D for AdaMerging$_{+\mathrm{TAS}}$ and AdaMMS$^*$ respectively.

## 4.2 RESULTS AND MAIN OBSERVATIONS

In Tab. 1, we present a summary of our extensive benchmarking results evaluated under the ablation protocol described in Sec. 4.1. Our key findings are summarized below.

**(i) Model merging is effective even in this challenging multi-task setting.** Despite the heterogeneity of tasks and the fact that models were fine-tuned independently using different protocols, several merging methods achieve strong results. To the best of our knowledge, this is the first study demonstrating that model merging can work well beyond image classification and across such a diverse set of vision tasks.

---

[1] https://github.com/naver/dune

| | ADE20k mIoU (↑) | NYUd rmse (↓) | Bedlam pa-pve (↓) | MapFree AUC (↑) | Normalized Performance | Merging cost |
|---|---|---|---|---|---|---|
| Base model | 46.1 | 0.382 | 61.2 | 0.924 | 0.821 | – |
| Fine-tuned[§] | **61.3** | 0.295 | 48.3 | 0.952 | – | – |
| **Closed-form** | | | | | | |
| Model average | 48.1 | 0.327 | 57.6 | 0.937 | 0.877 | <**30sec** |
| NormAvg | 48.5 | 0.317 | 59.5 | 0.932 | 0.878 | <**30sec** |
| MetaGPT | 31.9 | 0.496 | **42.2** | 0.891 | 0.799 | <**30sec** |
| **Hyperparameter-based** (*best results after exhaustive hyperparameter search*) | | | | | | |
| TSV-lossy | 41.6 | 0.467 | 68.6 | 0.912 | 0.743 | *>14 days*[†] |
| TIES | 43.7 | 0.414 | 62.6 | 0.926 | 0.792 | *>16 days*[†] |
| TSV-Lossless | 46.6 | 0.354 | 75.4 | 0.933 | 0.804 | *>14 days*[†] |
| Consensus | 47.8 | 0.357 | 63.6 | 0.937 | 0.837 | *>16 days*[†] |
| Breadcrumbs | 46.5 | 0.354 | 58.9 | 0.936 | 0.849 | *>13 days*[†] |
| Lines+Consensus | 48.1 | 0.352 | 60.4 | 0.938 | 0.852 | *>16 days*[†] |
| TA | 46.4 | 0.330 | 53.9 | 0.951 | 0.886 | *>14 days*[†] |
| Lines+TA | 47.5 | 0.325 | 55.9 | 0.948 | 0.886 | *>14 days*[†] |
| STAR | 48.0 | 0.327 | 54.6 | 0.949 | 0.892 | *>16 days*[†] |
| **Proxy-based** (*selecting best hyperparameters based on an efficient proxy*) | | | | | | |
| TA$_{+TAS}$ | 47.9 | 0.325 | 56.1 | 0.943 | 0.885 | <4min |
| STAR$_{+TAS}$ | 48.0 | 0.331 | 56.2 | 0.946 | 0.882 | <5min |
| AdaMMS* | 46.4 | 0.330 | 53.9 | 0.951 | 0.886 | <5min |
| AdaMerging $_{+TAS}$ | 51.0 | **0.294** | 59.4 | **0.956** | **0.913** | <30min |

Table 1: **Comparative results on the DUNE benchmark** using our ablation protocol. The best possible result is presented for the hyperparameter based methods, while all proxy-based methods needed some form of generalization to be applied here. Merging cost is calculated as the total GPU time (wall-clock time × number of GPUs). [§] Fine-tuned does not correspond to a single model but to the performance of the individual fine-tuned models per task. [†] For hyperparameter methods, cost is calculated for evaluating hyperparameters sequentially. *NormAvg* is a baseline which rescales task vectors to the smallest norm before averaging (see Sec. 4.3).

**(ii) Hyperparameter-based methods are prohibitively expensive.** These methods require training task-specific heads for every merged model during hyperparameter selection, leading to evaluation costs that are orders of magnitude higher than alternative approaches. This makes them impractical for real-world scenarios with limited computational resources. See also Fig. 2.

**(iii) Proxy-based methods are efficient and competitive.** Methods guided by TAS or AdaMMS* achieve performance close to the best hyperparameter-based methods, while being many orders of magnitude faster. In particular, AdaMerging$_{+TAS}$ achieves the best overall performance, while remaining significantly more efficient.

**(iv) Simple baselines remain strong.** Surprisingly, straightforward methods like Task Arithmetic (TA) and even plain weight averaging perform among the best, outperforming several recent techniques that were successful for CLIP-based classification. This observation echoes findings from the continual learning literature (Dziadzio et al., 2025; Sokar et al., 2025) and deserves further investigation. In the following section, we analyze the distribution of task vector norms and show that *large imbalances between task vectors may underlie this behavior.*

**Improving multi-task vision models via merging.** In Tab. 2, we evaluate merging-based models with the *long fine-tuning* protocol (see Sec. 4.1) introduced in Sarıyıldız et al. (2025) We find that all three models using TAS consistently outperform the base model across all tasks. We also ablate model distillation as an alternative to model merging, *i.e.* using the four fine-tuned encoders as teachers for distilling a student encoder, either initialized from scratch or from DUNE (See Appendix F). Despite spending considerably more compute, distillation underperforms arithmetic-based merging in this setting. We find this result surprising and worth exploring in future work.

| | ADE20k mIoU (↑) | NYUd rmse (↓) | Bedlam pa-pve (↓) | MapFree AUC (↑) | Normalized Performance | Merging cost |
|---|---|---|---|---|---|---|
| DUNE (CVPR25)[†] | 45.6 | 0.330 | 62.5 | 94.7 | 0.929 | – |
| **Distillation-based** | | | | | | |
| Distill (scratch) | 42.3 | 0.340 | 68.0 | 93.2 | 0.884 | ∼6 days |
| Distill (base) | 42.7 | 0.326 | 62.9 | 93.8 | 0.914 | ∼2 days |
| **Merging-based** | | | | | | |
| TA$_{+TAS}$ | 46.6 | 0.315 | 59.7 | 95.2 | 0.956 | <**5 min** |
| STAR$_{+TAS}$ | 46.6 | 0.312 | 59.3 | **95.7** | 0.962 | <**5 min** |
| AdaMerging$_{+TAS}$ | 49.3 | **0.300** | 62.9 | 95.2 | **0.971** | <30 min |

Table 2: **Improving multi-task vision models via merging.** We report results on the DUNE benchmark under the evaluation protocol introduced in the paper (Sarıyıldız et al., 2025). We also report results using distillation as an alternative to merging, *i.e.* using the four fine-tuned models as teachers, when initializing the student from scratch or from the DUNE model. [†] denotes reproducing the public DUNE model in our evaluation setup (note that we were unable to reproduce the 56 PA-PVE reported on Bedlam and we use a DPT decoder for depth estimation).

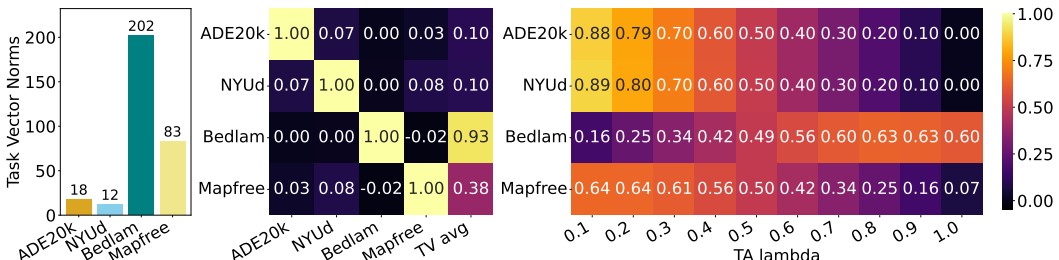

Figure 5: **Left:** Task vector norms for each task. **Middle:** Cosine Similarity between task vectors and between task vectors and their sum. **Right:** Normalized similarity (in weight space) between merged models and each task vector for different lambdas.

x

## 4.3 TASK VECTOR ANALYSIS

We now investigate why several recent merging methods underperform compared to simple baselines such as *Task Arithmetic* (TA) in our setting. Our analysis reveals that *task vector norms are highly imbalanced*, introducing implicit biases and trade-offs between tasks. This may explain why more sophisticated methods struggle in this multi-task vision setting.

**Task vector norms are highly unbalanced.** Fig. 5 (left) shows the relative $\ell_2$ norms of the task vectors for each task. We observe a wide variation in norms, with Bedlam having the largest and ADE20k and NYUd the smallest. We hypothesize that this imbalance arises from the diverse nature of the tasks and the differing fine-tuning protocols used. In contrast, as shown in Appendix G, the task vectors from the CLIP benchmark (downloaded from Gargiulo et al. (2025)) are much more comparable in magnitude.

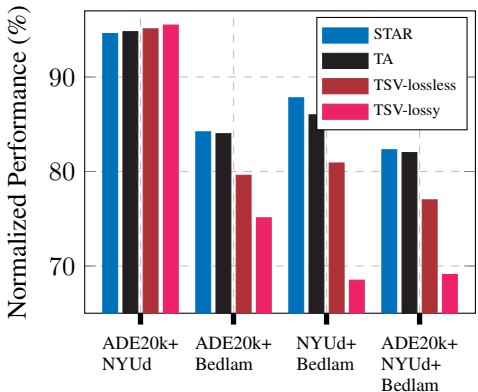

Figure 6: **Performance *vs.* task subsets.** When task vector norms are balanced (leftmost) we recover merging method rankings reported in the CLIP benchmark.

Fig. 5 (middle) shows that this norm imbalance biases the average task vector heavily towards Bedlam, leading to lower similarity with the other task vectors, especially ADE20k and NYUd. This introduces a clear trade-off: as shown in Fig. 5 (right), increasing the merge weight $\lambda$ pulls the merged model closer to Bedlam, but pushes it further from other tasks. This suggests that no single weight configuration can satisfy all tasks equally well.

**Task subset experiments.** To further explore the impact of task vector imbalance, we conduct merging experiments on task subsets. We consider the following 4 subsets (some with balanced norms): {ADE20k, NYUd}, {ADE20k, Bedlam}, {NYUd, Bedlam}, and {ADE20k, NYUd, Bedlam}. We exclude MapFree due to its high evaluation cost. Results are shown in Fig. 6.

We find that all methods achieve their best normalized performance when merging {ADE20k, NYUd}, the two tasks with the most similar task vector norms. Performance consistently degrades when Bedlam is included, indicating increased "merging difficulty." Beyond the overall performance drop, we also observe a complete reversal in the relative ranking of methods once Bedlam is added, closely mirroring the trends observed in Tab. 1. This highlights that some merging methods are more sensitive to task imbalance than others.

**Some merging methods implicitly expect balanced task vector norms.** In the CLIP benchmark, all tasks are classification problems and follow the same fine-tuning protocol. As a result, task vectors have similar norms (see also Fig. 12). Although not stated explicitly, we argue many merging methods seem to rely on this assumption—leading to performance degradation when it is violated. For instance, TIES performs weight pruning on task vectors based on magnitude and sign conflicts. Sign pruning involves comparing each task vector's sign to the sign of the average task vector, pruning mismatched signs. As shown in Fig. 5, the average vector is heavily biased toward Bedlam, so the sign pruning step disproportionately favors that task. Indeed, in Tab. 4, we show that increasing the pruning ratio improves Bedlam performance while hurting performance on other tasks.

**Normalizing task vectors.** If all tasks are to be weighted equally, a natural baseline is to normalize all task vectors before merging. We call this baseline **NormAvg**, where each task vector is rescaled to the smallest norm before averaging. As shown in Fig. 7, this produces a more balanced cosine similarity across tasks. In Tab. 1, we observe that NormAvg improves performance on low-norm tasks like ADE20k and NYUd compared to vanilla averaging, but degrades performance on Bedlam and MapFree. Ultimately, both strategies result in similar normalized performance over all tasks.

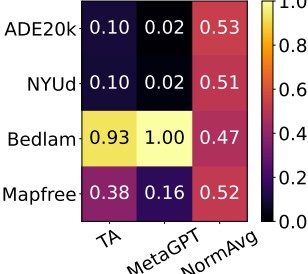

Figure 7: **Cosine similarity** of task vectors and models.

MetaGPT merges models using weights proportional to the norms of their task vectors. As a result, it further amplifies the biases introduced by the imbalance in task vector norms. Fig. 7 confirms this, showing that MetaGPT produces the most Bedlam-aligned model. Interestingly, despite its low overall normalized performance, MetaGPT achieves the best result on Bedlam, even surpassing its fine-tuned counterpart. We hypothesize that this may arise from synergies with other task vectors.

## 5 Conclusions

In this paper, we explore model merging beyond CLIP-based classification by evaluating existing methods on a benchmark of diverse vision tasks. Our study uncovers several findings: *i)* model merging, even in its simplest form (weight averaging) is effective even in this challenging setting. *ii)* Several recent merging methods underperform simple baselines. *iii)* When downstream tasks rely on different trainable decoders, the evaluation of merged encoders is very expensive.

To make hyperparameter-based methods affordable, we propose Task alignment score or TAS, a performance proxy that can also be used to generalize learning-based methods like AdaMerging. In fact, *any* merging method can leverage TAS for selecting hyperparameters at a fraction of the cost.

We also analyze why several recent merging methods fail in this setting. By comparing the variation of task vector norms across models fine-tuned for diverse vision tasks versus the variation of these norms across classification datasets, we find that the former exhibit *significantly higher imbalance*. This imbalance leads to large changes in the relative ranking of merging methods compared to prior work. Notably, when high-norm tasks are removed, the original ranking is largely restored—suggesting that high task vector norms might be an important factor in the poor performance observed in our (more challenging) setting.

We also observe that naively rescaling task vectors is insufficient, and that more nuanced weighting strategies are needed. In this context, adaptive methods like AdaMerging, which learn optimal weights per task, are better suited for robust merging across imbalanced task distributions.

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

# APPENDIX

## CONTENTS

## A THE USE OF LLMS IN THIS PAPER

LLMs were used to aid in the writing of the paper, suggesting structure of contents and reviewing the writing for clarity and typos. There is however, no text written only by an LLM directly. LLMs were also used to help in code generation for the plots and tables.

## B ADDITIONAL DETAILS ON THE EXPERIMENTAL PROTOCOL

In Sec. 4.1 we provide an overview of the evaluation protocols used in our experiments. In this section we provide additional details.

**Fine-tuning settings and costs.** We follow two different setups when fine-tuning decoders and evaluating models: the *ablation protocol*, where we limit the computational budget per fine-tuning; and the *long fine-tuning protocol*. For the long fine-tuning protocol we follow the same settings as described in Sarıyıldız et al. (2025):

- ADE20k: We finetune a linear head as in Oquab et al. (2024) for 80k iterations.
- NYUd: Similarly, we follow Oquab et al. (2024), and finetune for 48k iterations.
- Bedlam: We follow Baradel et al. (2024) and finetune for 200k iterations.
- MapFree: We follow Leroy et al. (2024) and finetune for 10 epochs on several datasets with 3D correspondences.

Of the above, MapFree fine-tuning is by far the most expensive as it involves the largest decoder. For this protocol, MapFree fine-tuning requires 4 H100s and takes around 4 days to complete. Other tasks take less than 24 hours on a single H100.

For the shorter *ablation protocol* we do as follows:

- ADE20k: We finetune a linear head as in Oquab et al. (2024) for 40k iterations.
- NYUd: Similarly, we follow Oquab et al. (2024), and finetune for 24k iterations.

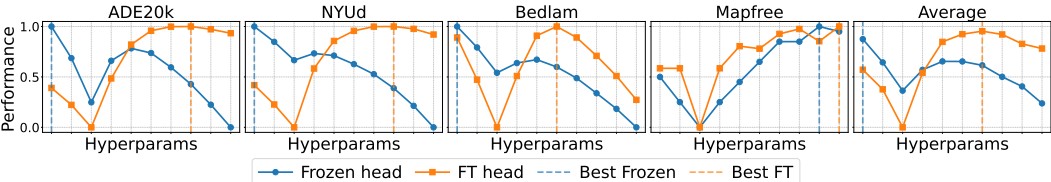

Figure 8: **Frozen decoders ablation**. We show the normalized performance when evaluating merged models with frozen decoders vs those same models when decoders are trained. We observe that frozen decoders performance is not a good proxy for trained decoders performance.

- Bedlam: We follow Baradel et al. (2024) and finetune for 60k iterations.
- MapFree: We follow Leroy et al. (2024) and finetune for 3 epochs only on the MapFree dataset (which was part of the training dataset mix used in Leroy et al. (2024)).

With this ablation protocol, the MapFree fine-tuning can be done in about 5 hours with 4 H100s and the other tasks take less than 12 hours with a single GPU.

Interestingly, in the original work of Sarıyıldız et al. (2025), authors observe that for ADE20k, keeping the distillation projector for the DINOv2 teacher (*i.e.*, a small teacher-specific head that projects student encoder features into the teacher feature space) improves performance significantly. We find that keeping the distillation projector improves performance, *even if the teacher projector is frozen*, *i.e.*, not part of the per-task fine-tuning or merging. As in Sarıyıldız et al. (2025), for other tasks, we do not see improvements when keeping the teacher projectors.

**Merging methods.** Unless otherwise stated, methods have been implemented following the settings described in their respective papers and following the public code implementations whenever available. However, for MetaGPT, instead of looking at the task vector norms from the vectorized full models, we found that computing the task vector norms per-layer (*i.e.* looking at the task-vector parameters at each layer independently), and consequently, finding a lambda per layer, led to better results. In a nutshell, Zhou et al. (2024) suggested to have

$$\lambda_t = \frac{||\theta_t - \theta_0||_2}{\sum_{j=1}^{T} ||\theta_j - \theta_0||_2}.$$  (3)

In this work, we consider MetaGPT per-layer which defines:

$$\lambda_t^l = \frac{||\theta_t^l - \theta_0^l||_2}{\sum_{j=1}^{T} ||\theta_j^l - \theta_0^l||_2},$$  (4)

where the superscript $l$ denotes the weights corresponding to layer $l$. Symmetrically, for NormAvg, we can normalize the norms for the full model or per-layer. In Tab. 4 we find that looking at the norms per-layer works slightly better than at the full model level, hence in the paper we only consider the per-layer versions.

## C   MERGING WITH FROZEN DECODERS

One option to alleviate the cost of hyperparameter selection is to mimic zero-shot classification evaluations and to keep any existing decoders from the fine-tuned models. However, this would require fine-tuning in two stages, – *i.e.* first fine-tune decoders on top of frozen base model and then fine-tune the base model with frozen decoders – but this would limit the applicability of model merging "out-of-the-box" as this fine-tuning procedure is non-standard. Moreover, we show in Fig. 8 that the selected hyperparameters when decoders are frozen are not a good proxy of the best hyperparameters if decoders are fine-tuned.

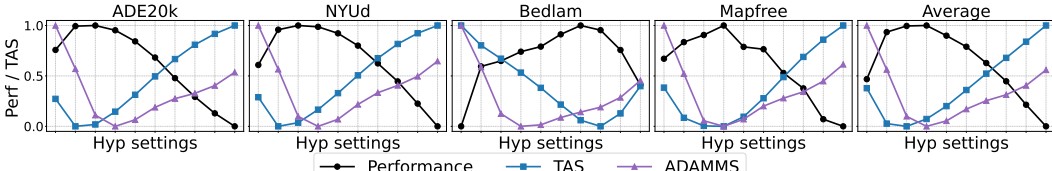

Figure 9: **AdaMMS* and TAS vs performance.** We found AdaMMS* to be agnostic to the input data domain (curves for all datasets are very similar), which leads to a bad approximation of the per-task performance, although, it seems to correlate reasonably well with the average performance. We hypothesize that the poor approximation of per-task performance is due to the lack of task-specific information (only merged models are compared between each-other).

## D    FURTHER DETAILS AND ABLATIONS OF ADAMMS*

As discussed in Sec. 2, AdaMMS was designed for generative models and relies on the comparison of model generated outputs for adjacent values of a **single** blending hyperparameter, named $\alpha$ in the original work. Let us denote $\alpha_0, \alpha_1, \ldots, \alpha_n$ a list of hyperparameter values in ascending order, *i.e.* $\alpha_i < \alpha_{i+1}$, then for a given $\alpha_i$ AdaMMS is defined as:

$$\text{AdaMMS}_{\alpha_i} = \frac{d(\alpha_{i-1}, \alpha_i) + d(\alpha_i, \alpha_{i+1})}{2},\tag{5}$$

where $d(\alpha_{i-1}, \alpha_i)$ denotes the distance between the outputs generated by the models merged with hyperparameters $\alpha_{i-1}$ and $\alpha_i$ respectively.

While this has the advantage that it only depends on the merged models (*i.e.*, this scales well with the number of tasks), the fact that it relies on the comparison of adjacent hyperparameter values makes it unsuitable to compare between merging methods (*e.g.*, one may want to efficiently compare merging methods for a given set of fine-tuned models) and more importantly, it only works with a single hyperparameter, because when more than one hyperparameter is involved, the ordering becomes less obvious and the number of "adjacent" hyperparameter values increases very quickly. This makes it unsuitable as a proxy for the large majority of recent merging methods. We adapt AdaMMS (and name it AdaMMS*) to compare model's output features (as opposed to generations). AdaMMS* generalizes well to any downstream task, however it is still unsuitable to compare different methods or methods with more than one hyperparameter.

In Fig. 9, we show AdaMMS* and TAS *vs.* performance. Interestingly, AdaMMS* seems to be agnostic to the input data domain (curves for all datasets are very similar), which leads to a bad approximation of the per-task performance, although it seems to correlate reasonably well with the average performance. We hypothesize that the poor approximation of the per-task performance is due to the lack of task-specific information (only merged models are compared between each-other).

## E    FURTHER ABLATIONS OF THE TASK ALIGNMENT SCORE

In this section we ablate the number of batches needed to obtain a good estimate of TAS, as well as different distance metrics that can be used to compare features, see Eq. (2). Fig. 10 shows the normalized performance vs TAS when using either a single batch or 50 batches per task. Interestingly we find that TAS is extremely robust to the number of samples used *i.e.* using 1 or 50 batches yields almost the same curves. Hence, we can use very few samples, making it very efficient to compute. On the other hand, Fig. 11 shows the performance vs TAS for all methods when using different distance metrics, namely $\ell_1$, $\ell_2$ and cosine distance (1 - cosine similarity). We observe that TAS is very robust to the choice of distance as well, with the scaled curves of all distances (min-max to 0-1) being almost the same.

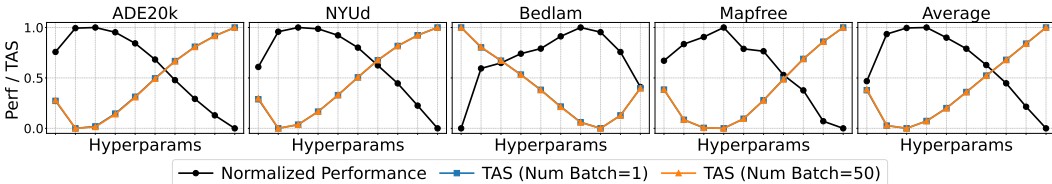

Figure 10: **Ablation of the number of batches for TAS.** We find that TAS is extremely robust to the number of samples used *i.e.* using 1 or 50 batches yields almost the same curves.

## F    DISTILLATION AS AN ALTERNATIVE TO MERGING

In Tab. 2 we report results using distillation as an alternative to merging, *i.e.* when using the four fine-tuned models as teachers for distillation. We report results when training from scratch or when initializing the student with the DUNE-448 model. For the experiment from scratch, we followed the original DUNE setup (Sarıyıldız et al., 2025) and first trained at resolution 336 for 100 epochs, followed by a second stage of distillation at resolution 448. When initializing from the DUNE-448 model, we only distilled at resolution 448. We first run a warm-up stage of 2 epochs where the encoder is frozen and only learn the TP projectors for the four teachers, followed by 50 epochs of full model training. We observe that distillation-based models do not improve over the base model.

**Longer distillation of the base model.** One could also consider extending the distillation of the DUNE model to more epochs. We experimented with continuing distillation from the publicly released DUNE-448 model, including a warm-up stage with projector-only training, but observed no additional performance gains.

## G    ANALYZING THE FINE-TUNED MODELS

**Task vector norms for the CLIP and DUNE benchmarks.** In Fig. 12 we plot the task vector norm for the 20 tasks of the CLIP benchmark, using the weights released by TSV (Gargiulo et al., 2025).[2] Since all models are ViT-Base, the absolute norm values are comparable to those in Fig. 5 for the DUNE benchmark. We see that even in the "furthest" case (EMNIST) the TV norm is much smaller than the "closest" of the four DUNE tasks (2.79 vs 12).

**Task vectors across layers.** We repeat the analysis from Figs. 5 and 12 when looking this time at the task vector norms per layer. For each layer, we flatten and concatenate all parameters in that layer, then calculate the $\ell_2$ norm between the base and fine-tuned models. In Fig. 13 we plot the task vector norms for each of the 12 layers of the ViT-Base model. We show the norms for each of the 4 tasks of the DUNE benchmark, as well as the average over all 20 CLIP benchmark tasks (with standard deviation shown as error bars).

## H    TAS AS A LOSS FOR ADAMERGING

Even though we present TAS as a proxy for final model performance, one can also look at the TAS objective from a differentiable optimization perspective and transform it into a *differentiable loss function*. In this section, we describe how to obtain such loss and how we have applied it to serve as the optimization objective in AdaMerging.

As discussed previously in Sec. 2, in AdaMerging the $\lambda$ merging coefficients are computed through parameter optimization. The authors proposed to use entropy minimization on unlabeled target samples as the surrogate objective for optimizing the merging coefficients, therefore making the method naturally applicable to classification tasks. In order to make it applicable to more generic tasks, we define a loss based on our general TAS structure.

---

[2]https://github.com/AntoAndGar/task_singular_vectors

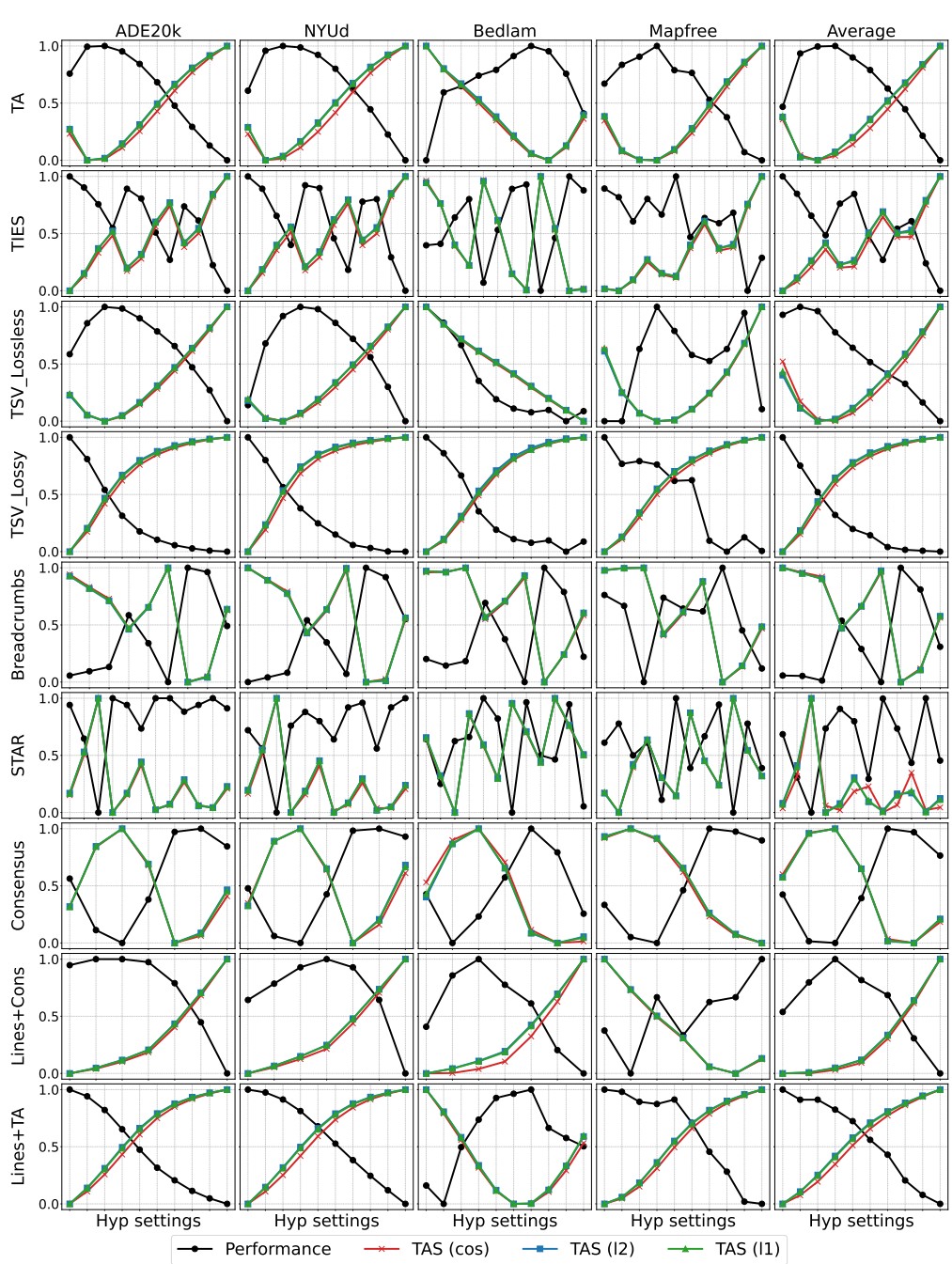

Figure 11: **Ablation of distance metrics for TAS.** TAS is very robust to the choice of distance, with the scaled curves for all distances (scaled with min-max to 0-1) being almost the same.

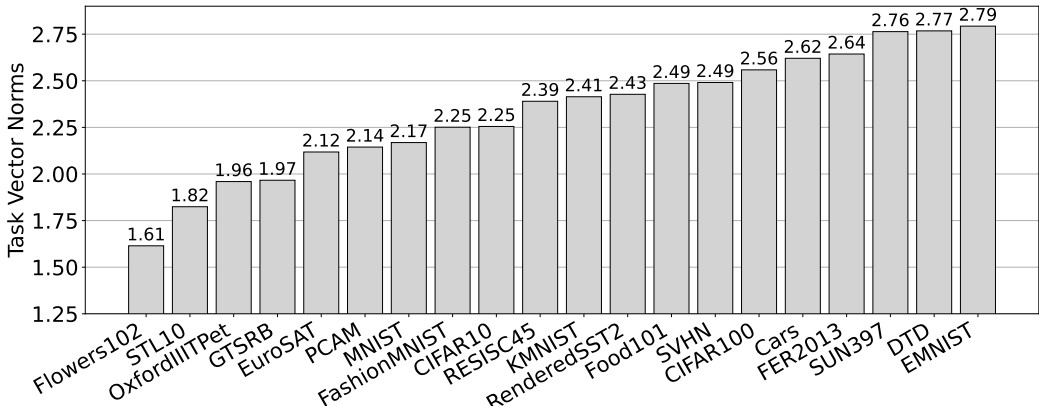

Figure 12: Task vector norms for the 20-task CLIP benchmark. For each task, we report the $\ell_2$ norm of the difference between its flattened parameter vector and that of the base model.

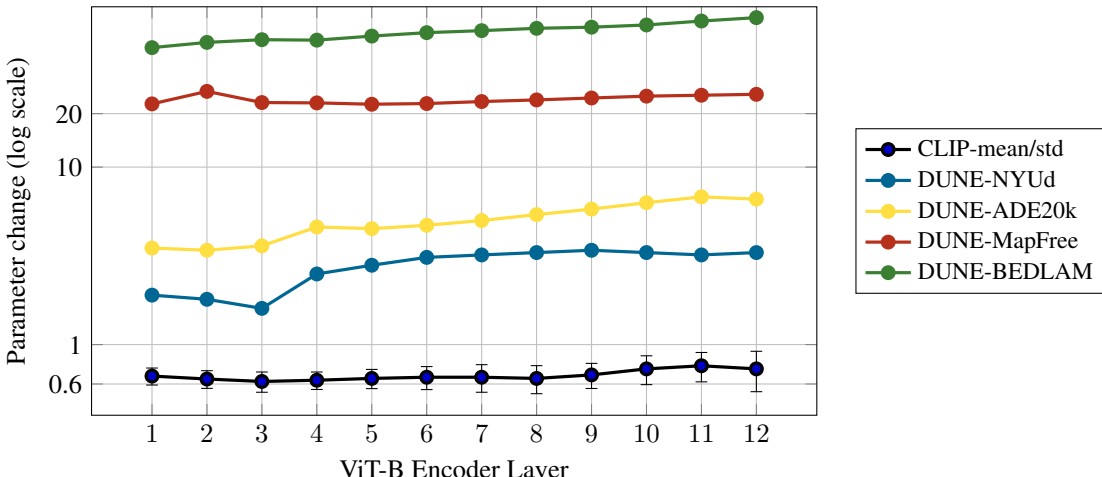

Figure 13: Task vector norms per layer for the CLIP and DUNE benchmarks. For CLIP we plot the average and standard deviation over all 20 tasks. See Tab. 3 for the full set of numbers.

## H.1 TASK ALIGNMENT SCORE AS A LOSS

Let $f(x; \theta_{merged})$ denote the encoder of the merged model, hereby referred as the *student*; and $f(x; \theta_t)$ the finetuned encoder of the task-$t$ as the *teacher*. Given an unlabeled dataset $D_t$ for task $t$, let $d : \mathcal{F} \times \mathcal{F} \to \mathbb{R}^+$ be a dissimilarity function defined over the feature vectors in the encoder space $\mathcal{F}$. We then define the *per-task alignment loss* as a function of the merging coefficients $\lambda$ as:

$$\mathcal{L}_{align}^{(t)}(\lambda) = \frac{1}{|D_t|} \sum_{x \in D_t} d(f(x; \theta_{merged}(\lambda)), f(x; \theta_t)), \tag{6}$$

where $\theta_{merged}(\lambda)$ is the merged encoder defined in Eq. 1, that in this case does not employ $\mu$ and $\phi$ is the identity. To obtain a single set of coefficients across all tasks, we then minimize the average task alignment loss:

$$\mathcal{L}_{align}(\lambda) = \frac{1}{T} \sum_{t=1}^{T} \mathcal{L}_{align}^{(t)}(\lambda). \tag{7}$$

| Model | Pos. | CLS | Conv. | Layer-1 | Layer-2 | Layer-3 | Layer-4 | Layer-5 | Layer-6 | Layer-7 | Layer-8 | Layer-9 | Layer-10 | Layer-11 | Layer-12 | Norm | Avg. |
|---|---|---|---|---|---|---|---|---|---|---|---|---|---|---|---|---|---|
| *CLIP benchmark* | | | | | | | | | | | | | | | | | |
| Cars | 0.1 | 0.0 | 0.1 | 0.7 | 0.8 | 0.7 | 0.7 | 0.7 | 0.7 | 0.6 | 0.6 | 0.7 | 0.8 | 0.9 | 1.0 | 0.0 | 0.6 |
| CIFAR10 | 0.1 | 0.0 | 0.1 | 0.7 | 0.6 | 0.6 | 0.6 | 0.6 | 0.7 | 0.7 | 0.6 | 0.7 | 0.7 | 0.7 | 0.6 | 0.0 | 0.5 |
| CIFAR100 | 0.1 | 0.0 | 0.1 | 0.7 | 0.6 | 0.6 | 0.6 | 0.7 | 0.7 | 0.7 | 0.8 | 0.8 | 0.9 | 0.9 | 0.8 | 0.0 | 0.6 |
| DTD | 0.1 | 0.0 | 0.1 | 0.7 | 0.7 | 0.7 | 0.7 | 0.8 | 0.8 | 0.8 | 0.8 | 0.8 | 0.9 | 0.9 | 0.9 | 0.0 | 0.6 |
| EMNIST | 0.1 | 0.0 | 0.1 | 0.8 | 0.7 | 0.7 | 0.7 | 0.7 | 0.8 | 0.8 | 0.8 | 0.8 | 0.8 | 0.9 | 1.0 | 0.1 | 0.6 |
| EuroSAT | 0.1 | 0.0 | 0.1 | 0.7 | 0.6 | 0.6 | 0.6 | 0.6 | 0.6 | 0.6 | 0.6 | 0.6 | 0.6 | 0.6 | 0.6 | 0.0 | 0.5 |
| FashionMNIST | 0.1 | 0.0 | 0.1 | 0.7 | 0.7 | 0.6 | 0.6 | 0.6 | 0.7 | 0.7 | 0.6 | 0.7 | 0.7 | 0.7 | 0.5 | 0.0 | 0.5 |
| FER2013 | 0.1 | 0.0 | 0.1 | 0.7 | 0.6 | 0.7 | 0.7 | 0.7 | 0.7 | 0.8 | 0.8 | 0.8 | 0.9 | 0.9 | 0.8 | 0.0 | 0.6 |
| Flowers102 | 0.1 | 0.0 | 0.1 | 0.5 | 0.5 | 0.4 | 0.5 | 0.5 | 0.4 | 0.4 | 0.4 | 0.5 | 0.5 | 0.5 | 0.5 | 0.0 | 0.4 |
| Food101 | 0.1 | 0.0 | 0.1 | 0.7 | 0.7 | 0.7 | 0.7 | 0.7 | 0.7 | 0.7 | 0.7 | 0.7 | 0.8 | 0.8 | 0.7 | 0.0 | 0.6 |
| GTSRB | 0.1 | 0.0 | 0.1 | 0.5 | 0.5 | 0.5 | 0.5 | 0.5 | 0.5 | 0.5 | 0.5 | 0.5 | 0.6 | 0.7 | 0.8 | 0.0 | 0.4 |
| KMNIST | 0.1 | 0.0 | 0.1 | 0.7 | 0.6 | 0.6 | 0.6 | 0.7 | 0.7 | 0.8 | 0.7 | 0.7 | 0.8 | 0.8 | 0.6 | 0.0 | 0.5 |
| MNIST | 0.1 | 0.0 | 0.1 | 0.6 | 0.6 | 0.6 | 0.6 | 0.6 | 0.6 | 0.6 | 0.6 | 0.6 | 0.6 | 0.7 | 0.9 | 0.0 | 0.5 |
| OxfordIIITPet | 0.1 | 0.0 | 0.1 | 0.6 | 0.6 | 0.6 | 0.6 | 0.6 | 0.6 | 0.5 | 0.5 | 0.5 | 0.6 | 0.6 | 0.5 | 0.0 | 0.4 |
| PCAM | 0.1 | 0.0 | 0.2 | 0.7 | 0.7 | 0.7 | 0.7 | 0.7 | 0.7 | 0.6 | 0.6 | 0.6 | 0.6 | 0.6 | 0.5 | 0.0 | 0.5 |
| RenderedSST2 | 0.1 | 0.0 | 0.2 | 0.6 | 0.6 | 0.6 | 0.6 | 0.6 | 0.6 | 0.6 | 0.6 | 0.8 | 0.9 | 0.9 | 0.8 | 0.0 | 0.5 |
| RESISC45 | 0.1 | 0.0 | 0.1 | 0.7 | 0.7 | 0.6 | 0.7 | 0.7 | 0.7 | 0.7 | 0.7 | 0.7 | 0.8 | 0.8 | 0.8 | 0.0 | 0.5 |
| STL10 | 0.1 | 0.0 | 0.1 | 0.6 | 0.6 | 0.5 | 0.5 | 0.5 | 0.5 | 0.5 | 0.5 | 0.5 | 0.5 | 0.5 | 0.4 | 0.0 | 0.4 |
| SUN397 | 0.1 | 0.0 | 0.1 | 0.7 | 0.7 | 0.7 | 0.7 | 0.7 | 0.7 | 0.8 | 0.8 | 0.8 | 0.9 | 1.0 | 1.0 | 0.0 | 0.6 |
| SVHN | 0.1 | 0.0 | 0.1 | 0.7 | 0.7 | 0.7 | 0.7 | 0.7 | 0.7 | 0.7 | 0.7 | 0.7 | 0.7 | 0.8 | 0.9 | 0.0 | 0.6 |
| *DUNE benchmark* | | | | | | | | | | | | | | | | | |
| NYU-d | 7.0 | 0.1 | 0.3 | 1.9 | 1.8 | 1.6 | 2.5 | 2.8 | 3.1 | 3.2 | 3.3 | 3.4 | 3.3 | 3.2 | 3.3 | 0.0 | 2.5 |
| ADE20k | 1.4 | 0.0 | 0.6 | 3.5 | 3.4 | 3.6 | 4.6 | 4.5 | 4.7 | 5.0 | 5.4 | 5.8 | 6.3 | 6.8 | 6.6 | 0.1 | 3.9 |
| MapFree | 6.9 | 0.2 | 2.4 | 22.7 | 26.7 | 23.1 | 23.0 | 22.6 | 22.8 | 23.4 | 23.9 | 24.5 | 25.1 | 25.4 | 25.7 | 0.0 | 18.6 |
| BEDLAM | 27.9 | 0.6 | 5.5 | 47.1 | 50.4 | 52.2 | 51.9 | 54.7 | 57.2 | 58.7 | 60.5 | 61.4 | 63.2 | 66.5 | 69.5 | 0.0 | 45.5 |

Table 3: Per-layer task vector norms for the CLIP and DUNE benchmarks with the ViT-Base architecture. *Pos.*: positional embeddings, *CLS*: the CLS token, *Conv.*: the first convolutional layer producing patch embeddings, *Layer-i*: the i-th Transformer block, *Norm*: the final LayerNorm. We omit the comparison of CLIP parameters (the pre-LayerNorm and post-projection layers) which do not have correspondences in the DUNE-based ViT models.

It is worth emphasizing the distinction between the original use of TAS as a performance proxy and its role here as a loss function. While the proxy does not need to be differentiable, using it as an optimization objective requires $d(\cdot, \cdot)$ to be almost everywhere differentiable, so that gradients can be computed during training. We discuss this and other practical considerations below.

## H.2 PRACTICAL CONSIDERATIONS AND IMPLEMENTATION DETAILS

Although the use of this framework appears straightforward, its application requires careful attention to a few implementation details.

**Choice of dissimilarity function.** When using TAS as a proxy for performance, we observed little difference between setting $d(\cdot, \cdot)$ as the cosine dissimilarity or the $\ell_2$ distance. However, when used as a learning objective we found the cosine dissimilarity to be more stable across experiments. We therefore used $d(\cdot, \cdot) = 1 - \cos(\cdot, \cdot)$ throughout our experiments.

**Visual transformer tokens.** For the particular case of ViT-based encoders, we apply the alignment loss element-wise over tokens and then average. We do not average the tokens themselves. In addition, we consider only patch tokens and do not attempt to align CLS or register tokens.

**Normalization.** We apply normalization via an exponential moving average (EMA) of feature statistics maintained during training for the student and the teachers.

**Task-specific structure.** Feature-level alignment, unlike its use as a performance score, may have unintended effects. Task-specific encoders often allocate capacity to features that are only useful within their domain (*e.g.*, MultiHMR). For instance, models trained on data with strong spatial or semantic biases (such as datasets emphasizing specific human body parts for pose recovery) may produce ambiguous or noisy features in other regions. Our alignment loss does not differentiate between informative and less relevant dimensions, and may therefore encourage the student to replicate such task-specific artifacts. This observation corroborates with our experiments, in which we found our Bedlam teachers to be most challenging to approximate (see Fig. 14).

**Learning.** For optimizing the merging coefficients, we follow the same settings as in Yang et al. (2024). Specifically, we use the Adam optimizer (Kingma & Ba, 2014) with a learning rate of $1 \times 10^{-3}$, a batch size of 16, and run for 500 iterations. The merging coefficients $\lambda$ are initialized uniformly, *i.e.*, $\lambda = 1/T$ (*e.g.*, $1/4 = 0.25$ in our case with four tasks).

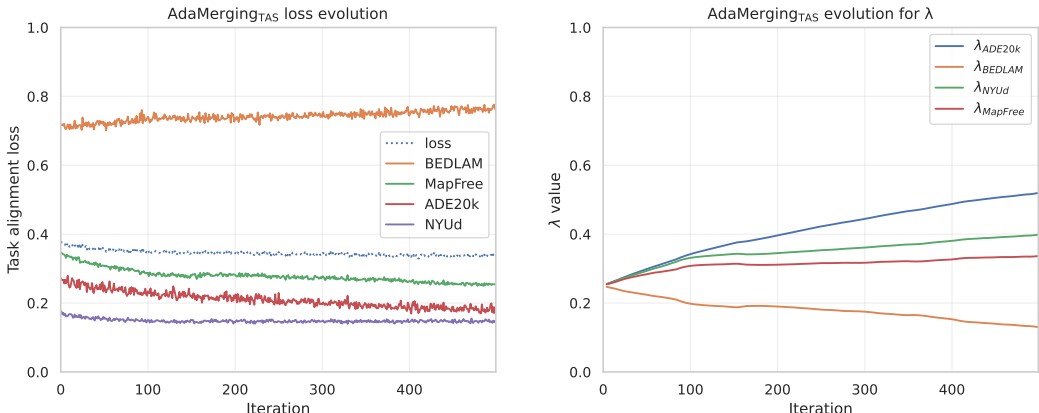

Figure 14: **Left:** Loss curves observed while training AdaMerging$_{+TAS}$. We can observe similar behavior between dense tasks (ADE20k, NYUd), and a different behavior in Bedlam. **Right:** $\lambda$ values for different tasks observed while training AdaMerging$_{+TAS}$. Again, we observe that similar tasks show a similar pattern, while more specific tasks (Bedlam) go a different path.

## I HYPERPARAMETER SEARCH RESULTS

Tab. 1 only shows the best hyperparameter setting per method for brevity, but in Tab. 4 we report the results of all the models evaluated during hyperparameter search for each method. Similarly as in Tab. 1, we use color shading to indicate relative performance compared to the base model. However, here we highlight in bold the best number per method, for each column. Note that the selected model for each method corresponds to the model with highest normalized performance.

| Method | Hyperparams | ADE20k mIoU (↑) | NYUv2d rmse (↓) | Bedlam pa-pve (↓) | Map-free AUC (↑) | Normalized Performance |
|---|---|---|---|---|---|---|
| TA | $\lambda$: 0.1 | 40.2 | 0.476 | 71.6 | 0.923 | 0.730 |
| TA | $\lambda$: 0.2 | 47.7 | 0.341 | 57.4 | 0.937 | 0.867 |
| TA | $\lambda$: 0.3 | **47.9** | **0.325** | 56.1 | 0.943 | 0.885 |
| TA | $\lambda$: 0.4 | 46.4 | 0.330 | 53.9 | **0.951** | **0.886** |
| TA | $\lambda$: 0.5 | 42.9 | 0.355 | 52.7 | 0.933 | 0.857 |
| TA | $\lambda$: 0.6 | 37.8 | 0.402 | 49.8 | 0.931 | 0.825 |
| TA | $\lambda$: 0.7 | 31.3 | 0.470 | **47.7** | 0.911 | 0.777 |
| TA | $\lambda$: 0.8 | 25.4 | 0.539 | 48.8 | 0.898 | 0.724 |
| TA | $\lambda$: 0.9 | 20.2 | 0.624 | 53.5 | 0.872 | 0.655 |
| TA | $\lambda$: 1.0 | 16.1 | 0.711 | 61.8 | 0.866 | 0.592 |
| TIES | $\lambda$: 0.8, K: 0.05 | **43.7** | **0.414** | 62.6 | 0.926 | **0.792** |
| TIES | $\lambda$: 0.8, K: 0.1 | 41.2 | 0.453 | 62.4 | 0.921 | 0.766 |
| TIES | $\lambda$: 0.8, K: 0.2 | 37.4 | 0.539 | 58.8 | 0.907 | 0.733 |
| TIES | $\lambda$: 0.8, K: 0.3 | 32.0 | 0.631 | 56.3 | 0.920 | 0.703 |
| TIES | $\lambda$: 1.0, K: 0.05 | 40.9 | 0.442 | 67.7 | 0.911 | 0.751 |
| TIES | $\lambda$: 1.0, K: 0.1 | 38.7 | 0.451 | 60.5 | **0.933** | 0.766 |
| TIES | $\lambda$: 1.0, K: 0.2 | 31.0 | 0.610 | 54.9 | 0.898 | 0.703 |
| TIES | $\lambda$: 1.0, K: 0.3 | 24.8 | 0.710 | 54.3 | 0.909 | 0.666 |
| TIES | $\lambda$: 1.2, K: 0.05 | 36.9 | 0.494 | 68.8 | 0.906 | 0.713 |
| TIES | $\lambda$: 1.2, K: 0.1 | 33.7 | 0.486 | 61.6 | 0.912 | 0.725 |
| TIES | $\lambda$: 1.2, K: 0.2 | 23.6 | 0.670 | **53.2** | 0.867 | 0.661 |
| TIES | $\lambda$: 1.2, K: 0.3 | 17.8 | 0.776 | 55.1 | 0.886 | 0.619 |
| TSV Lossless | $\lambda$: 0.6 | 44.7 | 0.381 | **68.6** | 0.933 | 0.797 |
| TSV Lossless | $\lambda$: 0.7 | 46.6 | 0.354 | 75.4 | 0.933 | **0.804** |
| TSV Lossless | $\lambda$: 0.8 | **47.6** | 0.342 | 85.0 | 0.945 | 0.800 |
| TSV Lossless | $\lambda$: 0.9 | 47.5 | **0.338** | 100 | **0.952** | 0.782 |
| TSV Lossless | $\lambda$: 1.0 | 46.9 | 0.339 | 108 | 0.948 | 0.769 |
| TSV Lossless | $\lambda$: 1.1 | 46.1 | 0.345 | 112 | 0.944 | 0.757 |
| TSV Lossless | $\lambda$: 1.2 | 45.2 | 0.352 | 114 | 0.943 | 0.748 |
| TSV Lossless | $\lambda$: 1.3 | 43.9 | 0.360 | 113 | 0.945 | 0.739 |
| TSV Lossless | $\lambda$: 1.4 | 42.5 | 0.373 | 118 | 0.951 | 0.723 |
| TSV Lossless | $\lambda$: 1.5 | 40.6 | 0.388 | 113 | 0.935 | 0.708 |
| TSV Lossy | $\lambda$: 0.6 | **41.6** | **0.467** | **68.6** | **0.912** | **0.743** |
| TSV Lossy | $\lambda$: 0.7 | 34.8 | 0.587 | 75.4 | 0.873 | 0.657 |
| TSV Lossy | $\lambda$: 0.8 | 25.2 | 0.729 | 85.0 | 0.877 | 0.576 |
| TSV Lossy | $\lambda$: 0.9 | 17.1 | 0.841 | 100 | 0.872 | 0.507 |

*Continued on next page*

| | Hyperparams | ADE20k mIoU (↑) | NYUv2d rmse (↓) | Bedlam pa-pve (↓) | Map-free AUC (↑) | Normalized Performance |
|---|---|---|---|---|---|---|
| TSV Lossy | $\lambda$: 1.0 | 12.2 | 0.920 | 108 | 0.848 | 0.464 |
| TSV Lossy | $\lambda$: 1.1 | 9.60 | 0.979 | 112 | 0.849 | 0.445 |
| TSV Lossy | $\lambda$: 1.2 | 7.90 | 1.03 | 114 | 0.760 | 0.409 |
| TSV Lossy | $\lambda$: 1.3 | 6.90 | 1.05 | 113 | 0.744 | 0.401 |
| TSV Lossy | $\lambda$: 1.4 | 6.20 | 1.07 | 118 | 0.765 | 0.398 |
| TSV Lossy | $\lambda$: 1.5 | 5.90 | 1.07 | 113 | 0.745 | 0.395 |
| Breadcrumbs | $\lambda$: 0.3, $\gamma$: 0.01, $\beta$: 0.1 | 41.5 | 0.452 | 67.2 | 0.926 | 0.755 |
| Breadcrumbs | $\lambda$: 0.3, $\gamma$: 0.01, $\beta$: 0.2 | 41.7 | 0.448 | 67.8 | 0.922 | 0.755 |
| Breadcrumbs | $\lambda$: 0.3, $\gamma$: 0.01, $\beta$: 0.3 | 41.9 | 0.444 | 67.4 | 0.894 | 0.751 |
| Breadcrumbs | $\lambda$: 0.5, $\gamma$: 0.01, $\beta$: 0.1 | 44.3 | 0.399 | 62.1 | 0.925 | 0.803 |
| Breadcrumbs | $\lambda$: 0.5, $\gamma$: 0.01, $\beta$: 0.2 | 43.0 | 0.418 | 65.4 | 0.921 | 0.778 |
| Breadcrumbs | $\lambda$: 0.5, $\gamma$: 0.01, $\beta$: 0.3 | 41.2 | 0.445 | 69.3 | 0.920 | 0.750 |
| Breadcrumbs | $\lambda$: 0.7, $\gamma$: 0.01, $\beta$: 0.1 | **46.5** | **0.354** | 58.9 | **0.936** | **0.849** |
| Breadcrumbs | $\lambda$: 0.7, $\gamma$: 0.01, $\beta$: 0.2 | 46.3 | 0.362 | 61.1 | 0.913 | 0.830 |
| Breadcrumbs | $\lambda$: 0.7, $\gamma$: 0.01, $\beta$: 0.3 | 43.8 | 0.398 | 67.0 | 0.899 | 0.780 |
| STAR | $\nu$: 0.6, $\lambda$: 0.8 | 47.8 | 0.332 | 56.3 | 0.946 | 0.880 |
| STAR | $\nu$: 0.6, $\lambda$: 1.0 | 46.8 | 0.336 | 58.5 | 0.949 | 0.866 |
| STAR | $\nu$: 0.6, $\lambda$: 1.2 | 44.6 | 0.350 | 56.4 | 0.944 | 0.855 |
| STAR | $\nu$: 0.8, $\lambda$: 0.8 | **48.0** | 0.331 | 56.2 | 0.946 | 0.882 |
| STAR | $\nu$: 0.8, $\lambda$: 1.0 | 47.8 | 0.328 | **54.3** | 0.937 | 0.888 |
| STAR | $\nu$: 0.8, $\lambda$: 1.2 | 47.1 | 0.330 | 55.3 | **0.953** | 0.884 |
| STAR | $\nu$: 0.95, $\lambda$: 0.8 | 47.8 | 0.336 | 57.3 | 0.935 | 0.871 |
| STAR | $\nu$: 0.95, $\lambda$: 1.0 | **48.0** | 0.327 | 54.6 | 0.949 | **0.892** |
| STAR | $\nu$: 0.95, $\lambda$: 1.2 | 47.7 | **0.325** | 59.6 | 0.942 | 0.871 |
| STAR | $\nu$: 0.9, $\lambda$: 0.8 | **48.0** | 0.334 | 59.9 | 0.942 | 0.866 |
| STAR | $\nu$: 0.9, $\lambda$: 1.0 | **48.0** | 0.327 | 54.5 | 0.947 | 0.892 |
| STAR | $\nu$: 0.9, $\lambda$: 1.2 | 47.6 | 0.326 | 57.1 | 0.952 | 0.882 |
| Consensus | $\lambda$: 0.1 | 44.9 | 0.415 | 68.3 | 0.911 | 0.777 |
| Consensus | $\lambda$: 0.2 | 41.7 | 0.463 | 71.8 | 0.900 | 0.734 |
| Consensus | $\lambda$: 0.3 | 40.9 | 0.470 | 69.9 | 0.898 | 0.732 |
| Consensus | $\lambda$: 0.4 | 43.6 | 0.421 | 67.1 | 0.916 | 0.773 |
| Consensus | $\lambda$: 0.6 | 47.8 | 0.357 | **63.6** | **0.937** | **0.837** |
| Consensus | $\lambda$: 0.8 | **48.0** | **0.355** | 65.3 | 0.936 | 0.834 |
| Consensus | $\lambda$: 1.0 | 46.9 | 0.363 | 69.7 | 0.933 | 0.813 |
| Lines+Consensus | $\lambda$: 0.1 | 47.9 | 0.356 | 66.2 | 0.931 | 0.829 |
| Lines+Consensus | $\lambda$: 0.2 | **48.1** | 0.354 | 61.8 | 0.922 | 0.842 |
| Lines+Consensus | $\lambda$: 0.3 | **48.1** | 0.352 | **60.4** | 0.938 | **0.852** |
| Lines+Consensus | $\lambda$: 0.4 | 48.0 | **0.351** | 62.6 | 0.930 | 0.843 |
| Lines+Consensus | $\lambda$: 0.6 | 47.3 | 0.352 | 64.2 | 0.937 | 0.837 |
| Lines+Consensus | $\lambda$: 0.8 | 46.0 | 0.356 | 68.2 | 0.938 | 0.818 |
| Lines+Consensus | $\lambda$: 1.0 | 44.3 | 0.365 | 70.2 | **0.946** | 0.803 |
| Lines+TA | $\lambda$: 0.1 | **47.5** | **0.325** | 55.9 | **0.948** | **0.886** |
| Lines+TA | $\lambda$: 0.2 | 45.8 | 0.332 | 58.1 | 0.946 | 0.865 |
| Lines+TA | $\lambda$: 0.3 | 42.3 | 0.349 | 51.3 | 0.937 | 0.865 |
| Lines+TA | $\lambda$: 0.4 | 37.4 | 0.377 | 48.0 | 0.935 | 0.845 |
| Lines+TA | $\lambda$: 0.5 | 32.2 | 0.414 | 45.4 | 0.939 | 0.822 |
| Lines+TA | $\lambda$: 0.6 | 27.6 | 0.456 | 44.9 | 0.918 | 0.784 |
| Lines+TA | $\lambda$: 0.7 | 24.4 | 0.496 | **44.4** | 0.892 | 0.754 |
| Lines+TA | $\lambda$: 0.8 | 21.7 | 0.534 | 49.0 | 0.874 | 0.703 |
| Lines+TA | $\lambda$: 0.9 | 19.8 | 0.569 | 50.2 | 0.847 | 0.673 |
| Lines+TA | $\lambda$: 1.0 | 18.4 | 0.602 | 51.2 | 0.845 | 0.655 |
| NormAvg | none | **48.2** | **0.332** | **63.6** | **0.931** | **0.853** |
| NormAvg per-layer | none | **48.5** | **0.317** | **59.5** | **0.932** | **0.878** |
| MetaGPT | none | **32.7** | **0.491** | **45.2** | **0.901** | **0.787** |
| MetaGPT per-layer | none | **31.9** | **0.496** | **42.2** | **0.891** | **0.799** |
| Model Average | none | **48.1** | **0.327** | **57.6** | **0.937** | **0.877** |
| AdaMMS* | none | **46.4** | **0.330** | **53.9** | **0.951** | **0.886** |
| TA+TAS | none | **47.9** | **0.325** | **56.1** | **0.943** | **0.885** |
| STAR+TAS | none | **48.0** | **0.331** | **56.2** | **0.946** | **0.882** |
| AdaMerging+TAS | none | **51.0** | **0.294** | **59.4** | **0.956** | **0.913** |

Table 4: **Performance on ablation protocol for all merged methods**, including all hyperparameter ablations. We use color shading to indicate relative performance compared to the base model. However, here we highlight in bold the best number of each column per method. Note that the selected model for each method corresponds to the model with highest normalized performance. Methods without hyperparameter search have all their numbers in bold.

