# OpenReview forum: "Model Merging beyond Image Classification: A Reality Check"
_ICLR.cc/2026/Conference — Submitted to ICLR 2026_

### Official Review · Reviewer_K4ar · 2025-10-29

**Soundness:** 3
**Presentation:** 3
**Contribution:** 2
**Rating:** 4
**Confidence:** 4

**Summary:**

This paper investigates the performance of weight-space arithmetic merging in more realistic and diverse visual tasks beyond CLIP image classification. The authors extend their research from the previously common CLIP-classification benchmark to the DUNE benchmark, which includes heterogeneous tasks such as semantic segmentation, depth estimation, and 3D human mesh reconstruction and relocalization. They point out that in such scenarios, evaluating the merging model requires fine-tuning the decoders for each task after merging, making hyperparameter search-based methods very expensive. To address this issue, the authors propose a Task Alignment Score (TAS)—calculating the distance between the merged encoder and each task-specific encoder in the feature space based on unlabeled data. This serves as a proxy for the final task performance, significantly reducing the cost of hyperparameter selection. The authors use TAS to guide various merging methods and achieve state-of-the-art results in multi-task merging.

**Strengths:**

1. The writing is quite easy to read and it was well-written
2. The motivation is important and pragmatic: shifting model-merging research from inexpensive and evaluable CLIP-classification scenarios to real-world multi-task vision settings that require training task-specific decoders is a natural and practically meaningful extension.
3. The method is generalizable. The authors demonstrate how to embed TAS into different methods such as AdaMerging and AdaMMS, thereby extending proxy methods that were originally only applicable to classification/generation to any vision task.

**Weaknesses:**

1. The number and diversity of benchmark tasks remain limited: Although DUNE itself includes 4 heterogeneous tasks, the conclusions are not yet fully conclusive on larger-scale tasks and on the robustness of the model.
2. The description of the sensitivity to TAS design details is insufficient: Although the Appendix mentions robustness to distance metrics and sample size, a more granular ablation on the impact of selecting which layer would be appreciated.
3. The time taken by the hyperparameter method in the table, which is >14 days, is an estimate based on sequential evaluation. However, in real-world deployments, multiple configurations can be evaluated in parallel, or a more intelligent search method can be used. Therefore, the assertion that it is "infeasible" needs further clarification.

**Questions:**

1. As a plug-and-play approach, this paper compares a limited number of methods. I wonder if this approach can lead to sustained improvements when applied to some of the latest state-of-the-art methods, such as wudi-merging[1], iso-merging[2].
2. TAS requires an unlabeled sample set Dt. Please explain whether this set of samples is part of the training set or comes from a domain different from the training distribution. How will the performance of TAS change if the sampling distribution is shifted?
3. Methods such as wudi merging do not require excessive hyperparameter searching. So, how does the performance of wudi merging[1] compare to that of tas under the settings in this paper?

[1] Whoever Started the Interference Should End It: Guiding Data-Free Model Merging via Task Vectors

[2] Task Left Behind: Isotropic Model Merging with Common and Task-Specific Subspaces

If the author can solve the question and the weakness well, i will raise my score.

---

### Official Review · Reviewer_CpmB · 2025-10-31

**Soundness:** 3
**Presentation:** 3
**Contribution:** 2
**Rating:** 4
**Confidence:** 4

**Summary:**

This paper studies model merging that combines encoders fine-tuned on individual downstream tasks into a single model, aiming to maintain performance while reducing deployment cost. It points out that prior advances in model merging have been largely confined to CLIP-based image classification, and extends the study to encoder-sharing yet decoder-specialized vision tasks, such as segmentation, depth estimation, 3D pose relocalization, and multi-human 3D mesh recovery.

**Strengths:**

1. The paper identifies the limitation of prior CLIP-focused evaluations of model merging and extends the assessment to heterogeneous DUNE tasks, covering both 2D and 3D tasks. This broadening of scope offers a more realistic tested for studying model merging.

2. The authors introduce TAS, a label-free and inference-only proxy that measures the feature distance between the merged encoder and each task’s fine-tuned encoder. Across the 4 DUNE tasks, TAS exhibits a strong negative correlation with downstream performance, making it a useful signal for hyperparameter selection.

3. Using TAS-guided selection, the workflow trains the decoders once under the chosen setting, which substantially reduces search cost on DUNE while surpassing the DUNE baseline in overall performance.

4. To investigate why merging that works well for CLIP classification is more challenging on DUNE, the paper quantifies task-vector magnitudes and directions across CLIP and DUNE, focusing larger and more imbalanced changes on DUNE that help explain the increased difficulty.

5. Ablations show that TAS is robust to the choice of distance metric (L1/L2/cosine produce similar curves) and that only a small number of unlabeled samples is sufficient to obtain a stable signal.

**Weaknesses:**

1. All core experiments and the TAS performance correlation analysis were evaluated only on the four DUNE tasks. Because the performance comparisons are conducted only on DUNE, to the best of my reading there are no results on other multi-task backbones or downstream tasks; consequently, the claim “We show that merging can improve state-of-the-art multi-task models” reads as DUNE-specific. Evaluation across additional backbones, decoders, and datasets would be necessary to support generalization.

2. To the best of my reading, while the paper claims to move beyond image classification, in the CLIP classification setting it only compares task-vector norms and distributions across downstream tasks and does not apply or evaluate TAS itself. It is true that TAS is not designed to solve CLIP classification itself; however, given the stated aim of extending beyond classification, a minimal evaluation on standard CLIP benchmarks, showing how TAS-guided hyperparameter selection performs relative to established methods, would have more convincingly supported the generality of the proposal.

3. This work stands in contrast to a key advantage emphasized in model merging methods such as Model Soups, Task Arithmetic, and TIES-Merging merging, which emphasize merging in weight space without additional training. In CLIP classification, the text encoder remains fixed, and the vision encoder is merged and evaluated, which enables training-free composition. By contrast, in the present setting the approach structurally depends on decoder retraining after merging.

4.  TAS is a useful label-free and inference-only proxy that reduces search cost by selecting promising hyperparameters, but it is not a merging algorithm that directly improves encoder-decoder alignment. Consequently, the work reads more as a cost reduction strategy within a merging and decoder training pipeline than as a conceptual advance in model merging itself. Framed this way, the extension beyond image classification lies primarily at the level of evaluation and practical tuning, which limits the scope of the claimed extension.

**Questions:**

1. Could you report additional experiments on at least one other multi-task backbone/decoder/dataset to test whether TAS performance correlation and TAS guided gains replicate beyond DUNE?

2. Although CLIP-based image classification tasks are not the primary focus, could you provide a minimal, compute-normalized evaluation on CLIP benchmarks to enable a precise comparison between TAS guided hyperparameter selection and existing model merging methods?

---

### Official Review · Reviewer_ue9Q · 2025-11-01

**Soundness:** 3
**Presentation:** 3
**Contribution:** 2
**Rating:** 4
**Confidence:** 3

**Summary:**

This paper investigates the problem of merging vision models fine-tuned on heterogeneous tasks, extending beyond the typical CLIP-based image classification scenario currently dominating the model merging literature. Through empirical studies on the DUNE benchmark across four diverse tasks, the authors identify computational bottlenecks in prior methods due to the need for retraining task-specific decoders after merging. To address this, the paper introduces the Task Alignment Score (TAS), a feature-based proxy for guiding hyperparameter selection efficiently in general multitask merging scenarios. The results show that TAS closely aligns with actual performance, making merging methods practical for more diverse task settings. Extensive experimental analysis highlights strengths and failure modes of various merging algorithms, with particular attention to task vector norm imbalance as a core challenge in multitask merging.

**Strengths:**

1. The paper tackles a clear gap by moving the conversation around model merging away from CLIP-style image classification toward true multitask settings with diverse output structures. This reveals both practical relevance and non-trivial technical issues that prior work sidestepped.
2. The introduction of TAS is a significant methodological addition, addressing the key bottleneck of hyperparameter search in costly non-classification scenarios. As demonstrated in Figure 3 and Figure 4, TAS shows a strong correlation with performance, especially in the presence of retrained, trainable decoders.
3. Through clear analysis of task vector norm imbalance, the paper challenges assumptions made by previous merging methods and provides evidenced explanations for why some sophisticated methods fail against basic weight averaging.

**Weaknesses:**

1. Despite arguing for generalization, experiments are restricted to DUNE’s four tasks. There is no sufficient evidence presented that TAS or the observed failures/successes generalize beyond this curated set.
2. A notable limitation is the absence of experiments on large-scale models. The scalability and effectiveness for contemporary foundation models, particularly LLMs and VLMs, have not been investigated.

**Questions:**

1. Are there regimes (task types, model types) where TAS would fail as a proxy?

---

### Official Review · Reviewer_6Kqn · 2025-11-01

**Soundness:** 2
**Presentation:** 1
**Contribution:** 2
**Rating:** 4
**Confidence:** 3

**Summary:**

The paper studies arithmetic model merging for computer vision beyond the usual CLIP-based classification setting. Using DUNE as a base encoder, the authors fine-tune four task-specific encoders and then merge only the encoders and re-train task-specific decoders for evaluation. They introduce a task-agnostic Task Alignment Score, which is the feature-space distance between a merged encoder and each task’s fine-tuned encoder. It is used as a cheap proxy to select merging hyperparameters or as a loss for learning-based merging. TAS correlates with downstream performance and enables efficient selection, making costly hyperparameter searching unnecessary. Empirically, several closed-form and proxy-based methods perform competitively, and experimental results show that AdaMerging+TAS improves over the base multi-task DUNE.

**Strengths:**

1. The setting in this paper more closely matches practical multi-task settings. In the manuscript, the authors Identified why exhaustive sweeps are infeasible in this regime.
2. TAS shows effectiveness and efficiency with empirical support. The design is label-free, task-agnostic and easy to follow. Merging cost results in Tab, 1 show that the method improves cost-effectiveness. The authors generalize AdaMerging with TAS and enable task-agnostic optimization of $\lambda$ .
3. Task-vector imbalance reveals some insights. The measurement shows norms are highly unbalanced across tasks, biasing average vectors. This work compares NormAvg and MetaGPT, then dissects how weighting by norms amplifies biases.

**Weaknesses:**

1. Limited breadth of tasks and datasets. Only four tasks from DUNE bench are studied and this limits external validity. Evidence of robustness is also mostly on the chosen benchmark. Providing cross-architecture evidence beyond the adopted size of ViT maybe more convinced.
2. The paper defines the merging goal. However, the evaluation protocol states that task-specific decoders are fine-tuned after the encoder is merged and frozen. This implies that $\mathcal{L}_t(\theta)$ is not a simple loss, but rather a nested optimization problem for the final expected performance. This is not made explicit, making the formal objective ambiguous to some extent.

**Questions:**

1. For merging cost calculation of hyperparameter methods, please provide more details.
2. which features of the adopted model are taken into account in $f(x;\theta)$ and being compared?
3. Other related questions please refer to the Weaknesses part.

---

### Official Review · Reviewer_PciQ · 2025-11-01

**Soundness:** 2
**Presentation:** 2
**Contribution:** 2
**Rating:** 2
**Confidence:** 5

**Summary:**

Most model-merging work has focused on simple CLIP-based classification setups. This paper evaluates merging in more challenging vision settings where each task has its own task-specific decoder. The authors introduce a Task Alignment Score (TAS) to accelerate hyperparameter search and to generalize proxy-based tuning methods. They also argue that disparities in task-vector norms explain why some “advanced” merging techniques underperform simpler baselines.

**Strengths:**

(1) Broader evaluation scope. Moving beyond CLIP-only classification to the DUNE setting makes the evaluation more realistic and practically relevant. Unsurprisingly, several prior approaches appear overfit to the CLIP regime.

(2) Insightful analysis. The study surfaces how task-vector norms and scaling coefficients affect merged-model performances and favour models with higher norm. This insight could help develop better merging and fine-tuning strategies for merging.

**Weaknesses:**

(1) Limited task diversity. While the DUNE setting is much more challenging than the CLIP Setting, the evaluation includes just four tasks and relies solely on this single setting for both evaluation and analysis, making generalisation uncertain. It would strengthen the paper to also include prior CLIP-style suites (e.g., up to 20 tasks).

(2) The idea of reducing the difference in feature representation between the merged model and the finetuned models represents the typical distillation objective and is not very novel.

(3) Attribution of norm effects. The work states that differences in the task-vector norm favour tasks with a higher norm, such as Bedlam, it is important to check if this is strictly due to the differences in norm or due to the nature of the task. To test this, one could regularise models to have a similar norm during training and check if such models merge in a more balanced manner.

(4) Modal coverage. Extending experiments to NLP—where model merging is widely used—would substantially increase the work’s relevance.

**Questions:**

(1) Feature definition in Eq. (2). Are the “features” the model’s output probabilities/logits or intermediate representations?

(2) Decoder retraining during tuning.  It is unclear why the decoders need to be trained for each hyperparameter selection during hyperparameter tuning, which leads to a significant increase in compute cost. It is typical to leave the layers not involved in merging as frozen during hyperparameter tuning.

**Details Of Ethics Concerns:**

No concerns.

---

### Author Response · Authors · 2025-11-27
**Response by authors**

We thank all five reviewers for constructive comments which will help make future versions of our manuscript much clearer. We were saddened to see that our main message did not come across, particularly along two core axes: 1) the reasoning behind the tasks we chose to tackle and 2) the nature of our novelty. We recognize that this was due to our the paper’s presentation, which we fully agree needs significant improvement.

**Task number and diversity**

Several reviewers mentioned that the number of tasks we tackle is small. Our focus, however, was not on the number of encoders we merge, but on __how different they are__. As we state in the supplementary material, even in the “furthest” classification case (EMNIST), the task vector norm is much smaller than the “closest” of the four DUNE tasks (2.79 vs. ~12 for NYUd, as shown in Figures 5 and 12).

A second major difference is that each task comes with additional modules. Unlike other all previous merging works, we therefore perform a partial merge.

We also want to note that merging for LLMs is similar to the classification case: no large, hard-to-train additional modules per task are present; LLMs are monolithic in the same way as classification encoders considered by previous computer vision work. We therefore consider the LLM use case beyond the scope of our paper.

Moreover, it is worth stating that unlike classification benchmarks in related merging works, where 20 datasets are considered “tasks” and where the upper bound of a jointly trained multi-task model is relatively easy to achieve with basic resources, we tackle a more realistic scenario with clear use cases, where joint training is infeasible without access to massive hardware compute. For this reason, we believe that using only the number of tasks as a proxy for problem difficulty is misleading.

**Novelty**

Unlike the classification case, where testing many hyperparameters is feasible, this is infeasible for our setting of vision models across diverse tasks. Each task comes with additional modules that cannot be merged, and each requires significant resources to finetune (the hardest, the MASt3R decoder training task, takes approximately 3–4 days on four strong GPUs).

Since we are interested in state-of-the-art performance across the multiple vision tasks we tackle, we show that achieving this requires finetuning the additional modules. Appendix C shows that a “plug-and-play” variant does not yield state-of-the-art results, making finetuning as a secondary step unavoidable.

To our knowledge, our paper is the first to attempt this very challenging case, and this is precisely where our novelty lies: we needed to identify a new metric to approximate final performance after finetuning. TAS turned out to (anti-)correlate with final performance surprisingly well (see Figure 4), and this is the main insight our paper contributes. What is more, TAS was necessary to extend merging methods like AdaMerge to tasks beyond classification.


We fully agree with the comment that we should be applying our method to more standard encoders such as DINOv2 besides DUNE. We have already tested this and observed that when fine-tuning and merging DINOv2 instead of DUNE for the same tasks, STAR+TAS achieves comparable performance, showing that our conclusions and insights are not tied to the DUNE encoder. Thank you for this suggestion.

---

### Meta-Review · Area_Chair_6TwZ · 2026-01-07

**Summary:**

This paper extends model merging research from CLIP-based classification to more realistic multi-task vision settings with task-specific decoders, using the DUNE benchmark. The authors introduce TAS (task alignment score), which is also a label-free proxy objective, enabling efficient hyper-parameter selection for merging methods like AdaMerging.

However, the reviewers share some common major concerns about this manuscripts:

1. All core results rely solely on DUNE's four tasks and one backbone; lacks broader evaluation (e.g., other encoders like DINOv2 beyond mentioned tests, CLIP benchmarks, or larger-scale/NLP settings) to support generalization claims.
2. TAS, while effective, resembles standard distillation proxies and is framed more as a tuning aid than a fundamental merging advance; the pipeline still requires decoder retraining.

The author response acknowledges presentation issues, and notes preliminary DINOv2 tests yield similar results. These address some concerns but do not fully resolve the reviewers' concerns.

Therefore, I would recommend rejection.

Additional note: The reference to "Le Yu et al. Ties-merging: Resolving interference when merging models. Proc. NeurIPS, 2023" appears incorrect in authorship—the actual TIES-Merging paper (NeurIPS 2023) is by Prateek Yadav et al. This is likely a BibTeX/citation error rather than a hallucination (paper exists, title/venue match closely), but please correct it in any revision.

**Reviewer Concerns:**

The authors partially address the reviewer concerns on task diversity (raised by all reviewers) and novelty (raised by reviewers PciQ, 6Kqn, CpmB). Specifically, the authors clarify focus on task heterogeneity (e.g., norm differences, partial merges) over quantity; note infeasibility of joint training and dissimilarity to LLMs/classification; mention preliminary DINOv2 tests showing similar results. For novelty of the manuscript, the authors emphasize TAS as key insight for hyperparam search and extending methods like AdaMerge; acknowledge presentation issues but defend as beyond standard distillation. Especially, the lack of broader evaluation and generalization are still outstanding.

**Reviewer Scores:**

I think the reviewers would make their scores unchanged.

---

### Decision · Program_Chairs · 2026-01-26

Reject